# Chemical Vapor Deposition of <110> Textured Diamond Film through Pre-Seeding by Diamond Nano-Sheets

**DOI:** 10.3390/ma15217776

**Published:** 2022-11-04

**Authors:** Guoyong Yang, Yunxiang Lu, Bo Wang, Yue Xia, Huanyi Chen, Hui Song, Jian Yi, Lifen Deng, Yuezhong Wang, He Li

**Affiliations:** Key Laboratory of Marine Materials and Related Technologies, Zhejiang Key Laboratory of Marine Materials and Protective Technologies, Ningbo Institute of Materials Technology and Engineering, Chinese Academy of Sciences, Ningbo 315201, China

**Keywords:** <110> textured diamond, diamond nano-sheet, pre-treatment, chemical vapor deposition

## Abstract

Diamond films prepared by chemical vapor deposition will exhibit different surface morphologies, which are determined by the texture and the structural perfection of the deposited diamond. In general, its surface morphology can be controlled by adjusting the deposition conditions. In the present work, <110> textured diamond film was deposited on single crystalline silicon through pre-seeding by diamond nanosheets, rather than controlling the deposition conditions. The employed diamond nano-sheets were prepared by cleavage along a plane, exhibiting good crystallinity. Before chemical vapor deposition, the as-prepared diamond nano-sheets were pre-seeded on the surface of single crystalline silicon as nucleation sites for diamond growth. SEM and XRD results show that the prepared diamond films have a <110> texture. FIB observation reveals that diamonds homogeneously grow on the pre-seeded diamond nano-sheets during chemical vapor deposition, achieving the diamond film with <110> texture. Our work provides a new strategy to prepare <110> textured diamond film.

## 1. Introduction

Properties of the diamond films prepared through chemical vapor deposition (CVD) vary according to their surface morphology [1,2,3,4,5]. Different surface morphologies of the diamond film can be obtained by varying the deposition conditions, including the concentration of methane, the chamber pressure, the deposition temperature, etc. [6,7]. Regarding the surface morphology, it mainly depends on two aspects, namely the shape of the deposited diamond grains and its corresponding orientation (texture) [8]. For the common deposition conditions, the shape of CVD diamond grain usually keeps tetrakaidecahedral, as shown in Figure 1a. In the tetrakaidecahedral diamond grain, there are 14 crystal planes consisting of six (100) planes and eight (111) planes. For <100> textured diamond, there is a (100) plane surrounded by four equivalent (111) planes. This will contribute to the low roughness of <100> textured diamond film. When transferring the orientation from <100> into <110>, two (100) planes and two (111) planes are presented. However, a (111) crystal plane is in the center of the tetrakaidecahedral diamond grain when observing along the <111> orientation. There are three (100) planes and three (111) planes around it.

In previous research, the diamond grain with different shapes, such as cubic, tetrakaidecahedral, and octahedral, were successfully prepared under different deposition conditions using CVD technology [9]. The shape of the deposited diamond grain was quantitatively studied in that work. It changes along the value of *R*, which represents the ratio of the growth rate along the <100> direction to that along the <111> direction. The corresponding textured diamond film has been theoretically investigated. According to the van der Drift competitive growth model, it is believed that diamond nucleate on the substrate randomly without orientation. Van der Drift explained the formation of the textured diamond film using the model of the survival of the fastest growing nucleus [10]. Under certain deposition conditions, there is a crystalline direction along which the fastest growth rate is achieved. Therefore, the nuclei in which the fastest direction is perpendicular to the substrate surface will prefer to grow bigger, resulting in a textured diamond film. R. E. Clausing et al. has also been testified the van der Drift model by experimental evidences [9]. The <110>, <111> and near <100> textured diamond films were prepared on (100) single crystalline silicon substrate through varying the gas compositions and deposition temperatures over a wide range using a hot-filament activated CVD equipment. From these above studies, it can be concluded that it is feasible to synthesize the diamond film with various shapes and orientations through controlling deposition conditions. However, it still is an interesting subject to directly grow diamond film with specific texture without probing the windows of the deposition parameters on different substrates.

It is reported that pre-treatment of the substrate, especially for pre-seeding, has a significant influence on the deposition of diamond prepared by CVD [10,11,12,13,14,15,16,17,18]. In this work, a novel strategy was proposed to prepare <110> textured diamond film. Prior to diamond deposition, the substrate was firstly pre-seeded by diamond nano-sheets with <110> orientation. During CVD process, the diamond will nucleate on the diamond nano-sheets which were pre-seeded on the surface of the substrate. Most of the diamond nuclei have <110> orientation or near <110> orientation, resulting in the synthesis of <100> textured diamond film.

## 2. Experimental Methods

The (100) single crystalline silicon wafers of 2 inch in diameter and 1 mm in thickness were chosen as the substrates. The silicon wafers were washed by ethyl alcohol and deionized water successively to clean up the purities on the wafer, followed by drying using the compressed nitrogen. CVD process was conducted on a homemade microwave plasma CVD (MPCVD) device, which has a maximum power of 6 kW. The employed power, deposition pressure, and the concentration of methane gas are 3.3 kW, 9 kPa, and 3% (H_2_: 400 sccm, CH_4_: 12 sccm), respectively.

Diamond nano-sheets were synthesized from synthetic diamond through cleavage plane crush separation, in which synthetic diamond successively underwent air steam milling and ball milling, followed by a sorting process, as described in the literature [19]. Prior to diamond deposition, the silicon wafers were pre-seeded using the diamond nano-sheets without other pre-treatment. The pre-seeded diamond nano-sheets were firstly dispersed in acetone. The diamond nano-sheet solution was dropped on the single crystalline silicon using a syringe. In order to evaporate the acetone, the silicon wafer was put on a heating plate of 60 ℃. During the evaporation of acetone, the diamond nano-sheets will be placed flat out on the silicon surface, realizing the pre-seeding by diamond nano-sheets.

Morphologies of diamond nano-sheets and the deposited diamond were observed using scanning electron microscopy (SEM, Regulus 8230, Hitachi, Japan) with an acceleration voltage of 5 kV. Micro-structure of the diamond nano-sheets was characterized by transmission electron microscopy (TEM, Talos F200X, Thermo scientific, USA) operating at an accelerating voltage of 200 kV. The thickness of diamond nano-sheets was measured by atomic force microscopy (AFM, Bruker Dimension ICON, Lincoln, USA) in tapping mode. Raman spectrum was performed on Raman Spectrometer (LabRAM Odyssey, Horiba, Lyon, France) with a laser wavelength of 532 nm. X-ray diffraction (XRD, Discover 8 Advance, Bruke, Billerica, USA) with a Cu target (λ = 0.154060 nm) was used to investigate the crystalline orientation. The cross-sectional sample of individual diamond grain was prepared and observed using focused ion beam microscopy (FIB, Helios-G4-CX, Thermo scientific, USA). Its operating voltage and tilt angle during FIB cutting were 30 kV and 52°, respectively.

## 3. Results and Discussion

Figure 2 displays the characterizations of raw diamond nano-sheets employed in pre-seeding prior to depositing diamond film on silicon substrate. From the XRD pattern shown in Figure 2a, three peaks can be clearly detected, which can be assigned to the (111), (220), and (311) planes in diamond structure, respectively. Regarding the intensities of these XRD peaks, the ratio of (111) peak to (220) peak is about 4, which is consistent to the standard XRD pattern of diamond. The XRD result demonstrates that there is no preferred orientation in the diamond nano-sheets. The diamond nano-sheet powders, which were used for collecting the XRD pattern, were compacted together before characterizing, resulting in random distribution without preferred orientation demonstrated in XRD pattern. In the Raman spectrum of raw diamond nano-sheets, the peak located at 1324 cm^−1^ is the characteristic Raman peak of diamond [20,21,22,23]. Compared to the standard Raman peak of diamond (1332 cm^−1^), it slightly shifted to the left, which is induced by the residual stress in the diamond nano-sheets produced in the synthesis process of the diamond nano-sheets. Another board peak located at about 1562 cm^−1^ can also be found in the Raman spectrum of raw diamond nano-sheets, which is the G peak for carbon materials [20,21,22,23]. The appearance of G peak verifies that, except for the diamond phase, there is a small amount of the graphite phase in the raw diamond nano-sheets. Morphology obtained by AFM shows that the thickness and the width of an individual diamond nano-sheet are about 10 and 60 nm, respectively, exhibiting its nature of two-dimensional material. Besides, the edge of the diamond nano-sheet is thinner than that in the center. Figure 2d,e are the TEM micrographs of raw diamond nano-sheets. As shown in Figure 2d, the size of diamond nano-sheets is about 100 nm, which agrees with the AFM results. In the enlarged view of TEM micrograph (Figure 2e), it can be seen that the nano-sheet has a perfect diamond structure, and the crystal spacing of (111) plane is 0.206 nm, illustrating that diamond structure remains stable during the synthesis of diamond nano-sheet [24].

After being pre-seeded by diamond nano-sheets, diamond was deposited on the substrate of single crystalline silicon. Figure 3a,b present the SEM images of the pre-seeded silicon surface at low and high magnifications, respectively. In order to observe the nucleation of diamond during CVD process, the silicon substrate was partially seeded by diamond nano-sheets. In Figure 3a, silicon surface was not fully covered by diamond nano-sheet, leading to the silicon of substrate being able to be observed. From the SEM image at high magnification, as displayed in Figure 3b, it can be found that the pre-seeded diamond nano-sheets flat out on the silicon surface, indicating that their (110) planes are parallel to the silicon surface. Figure 3c,d are the SEM images of the as-deposited diamond on the silicon surface during CVD. Due to the too short deposition duration, the deposited diamond did not form a complete film on the silicon substrate. However, the deposited diamond has a similar distribution on the silicon surface with the pre-seeded diamond nano-sheets prior to the CVD process. This demonstrates that the diamond during the CVD process epitaxially grew on the pre-seeded diamond nano-sheets. In the SEM image at high magnification (Figure 3d), it can be observed that the shape of the deposited diamond grains is tetrakaidecahedral. Moreover, the deposited diamond grains exhibit the same morphology with the model in Figure 1c, indicating that these diamond grains are mainly in (110) orientation. Three sharp peaks and one board peak can be found in the XRD pattern collected on the deposited diamond. The peaks located at 43.96°, 75.31°, and 91.44° can be assigned to the diamond (111), (220), and (311) planes, respectively. The board peak of 68.85° is induced by the (400) plane of silicon. In terms of peak intensity, it can be calculated that the ratio of the (111) plane to the (220) plane is 2, which is much smaller than that of the standard XRD pattern of diamond (I_111_/I_220_ = 4). The smaller ratio of the (111) plane to the (220) plane manifests that the deposited diamond has a strong (110) orientation, testifying the realization of the <110> textured diamond deposition through pre-seeding using diamond nano-sheets. In the Raman spectrum shown in Figure 3f, the peak located at 1335 cm^−1^ represents the signal of diamond, while the peak of 1529 cm^−1^ can be assigned to the graphite phase in the deposited diamond. Compared to the peak of 1529 cm^−1^, the characteristic peak of diamond is much sharper, illustrating the high crystallinity of the deposited diamond.

In order to investigate the growth mechanism of the <110> textured diamond on the diamond nano-sheets, the cross section of an individual diamond nucleus was cut and observed in FIB. In Figure 4a, both adjacent diamond nuclei are the same to the model shown in Figure 1c in shape, exhibiting (110) orientation. The bigger diamond nucleus was chosen to cut the cross-sectional sample. The corresponding cross-sectional SEM image is presented in Figure 4b after tilting the sample stage with an angle of 52°. In Figure 4b, two (111) planes on the top can be clearly detected. From the model in Figure 1c, it can be deduced that the side edges belong to (100) planes. In addition, there is an interface between the deposited diamond and the silicon substrate. However, the diamond nano-sheet is too thin to be observed clearly in FIB. Considering that the interface is perpendicular to the <110> direction, the interface can be defined as the (110) plane. Therefore, it can be speculated that the diamond epitaxially grew on the pre-seeded diamond nano-sheet, resulting in a <110> texture.

## 4. Conclusions

In this work, a novel approach of preparing <110> textured diamond was proposed using the CVD method. The deposition of the <110> textured diamond was achieved through pre-seeding diamond nano-sheets on the substrate of single crystalline silicon, which is different from controlling deposition conditions in previous studies. SEM results demonstrate that diamond will nucleate on the pre-seeding diamond nano-sheet, and the shape of the deposited diamond grains is tetrakaidecahedral. Combined with the SEM observation, XRD results manifest that the deposited diamond exhibits a <110> texture. In addition, the mechanism of the epitaxial growth of the <110> textured diamond has been investigated through the preparation and observation of a cross-sectional sample on an individual diamond grain.

## Figures and Tables

**Figure 1 materials-15-07776-f001:**
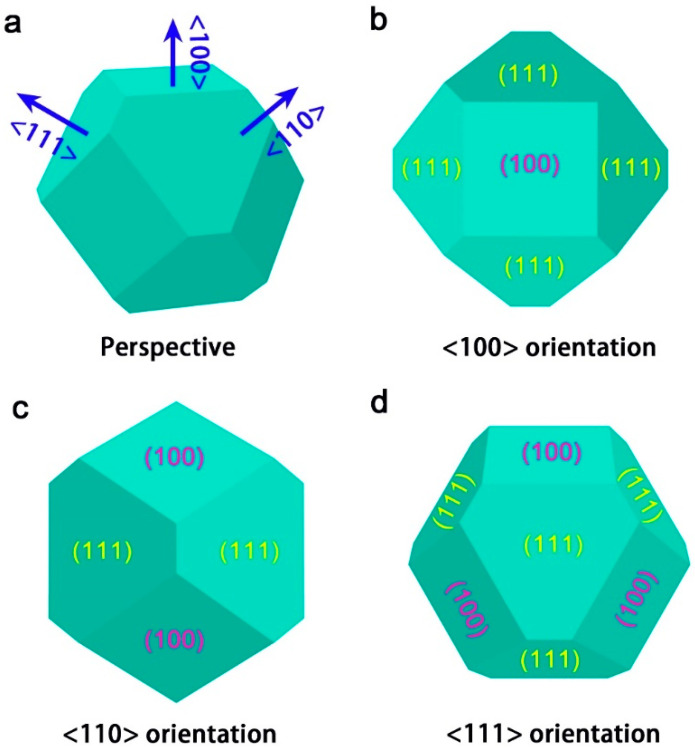
(**a**) Perspective drawing of the tetrakaidecahedral diamond grain. Crystalline faces shown in (**b**) <100>, (**c**) <110>, and (**d**) <111> orientations.

**Figure 2 materials-15-07776-f002:**
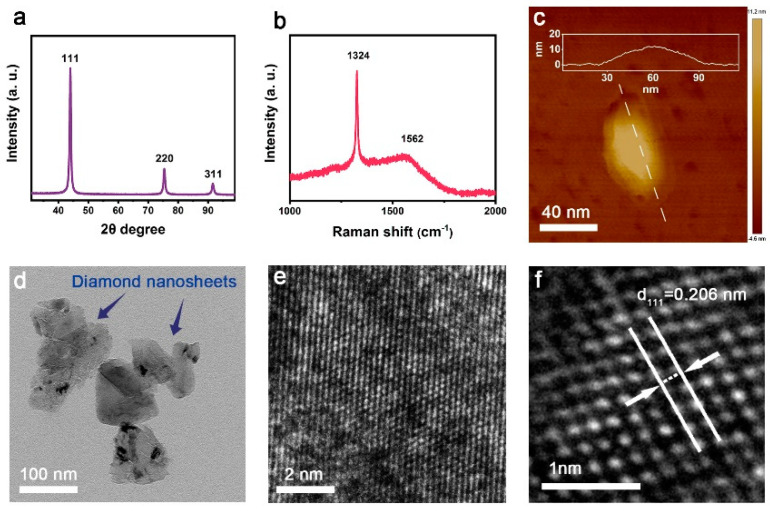
Characterizations of diamond nano-sheets. (**a**) XRD pattern, (**b**) Raman spectrum and (**c**) AFM morphology of diamond nano-sheets. TEM micrographs at (**d**) low and (**e**) high magnifications. (**f**) Enlarged view of the TEM micrograph in (**e**).

**Figure 3 materials-15-07776-f003:**
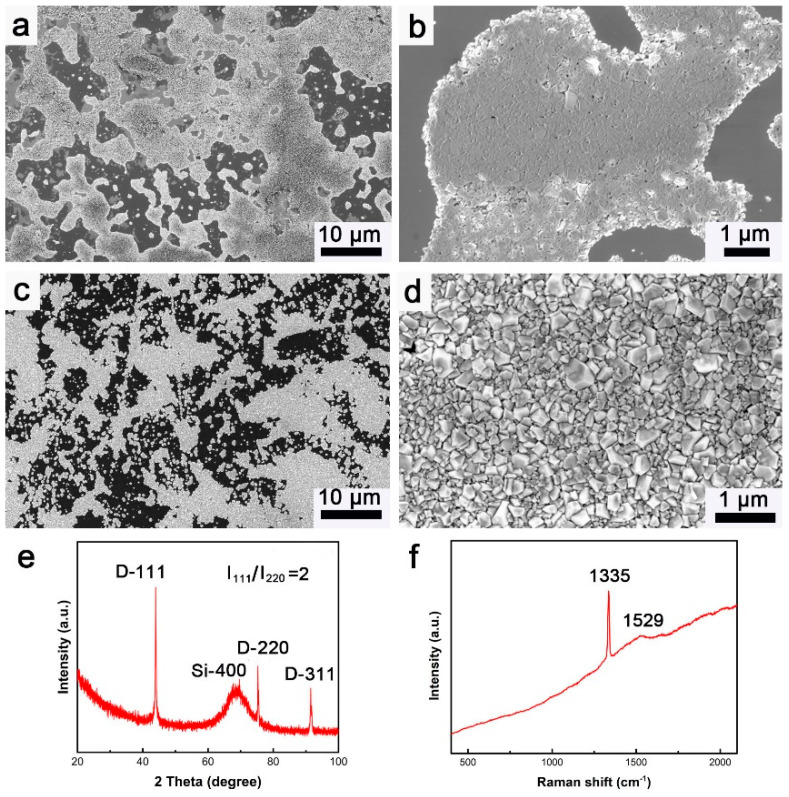
SEM images of the pre-seeded diamond nano-sheets on silicon surface at (**a**) low and (**b**) high magnifications; SEM images of the as-deposited diamond on the silicon surface at (**c**) low and (**d**) high magnification; (**e**) XRD pattern and (**f**) Raman spectrum of the as-deposited diamond on the silicon surface.

**Figure 4 materials-15-07776-f004:**
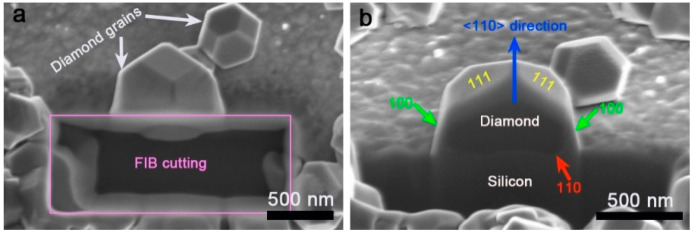
(**a**) Top view and (**b**) cross section of an individual diamond nucleus.

## Data Availability

Data available within the article.

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
