# Peer review of "Chemical Vapor Deposition of <110> Textured Diamond Film through Pre-Seeding by Diamond Nano-Sheets"

_materials, 2022, doi:10.3390/ma15217776_

Round 1

Reviewer 1 Report

see attached file

Author Response

Dear reviewer,

Thanks for your critical comments on our manuscript. Response on your and other reviewers' comments can be found in the enclosed document. Please feel free to comment on the revised version. We are glad to receive your suggestions again to further improve the quality of the manuscript.

Reviewer 2 Report

This manuscript is dedicated to the chemical vapor deposition of <110> textured diamond film taking advantage of a pre-seeding strategy of the silicon substrate by using diamond nanosheets. The work offers an original approach to controllably synthesize diamond films, with preferential <110> texture, without complex tuning of the experimental growth conditions. The advantages of this facile approach reveal an original and prospective topic of application of CVD, that currently attracts much research interest in several fields. The conclusions of the manuscript are well-supported by the data since the authors employed an array of adequate characterizations and detailed analysis (XRD, Raman, TEM).

Eventually, the insightful discussion of the results, as well as the very good figures, permit the readers to use the growth scheme of this manuscript as a guidance to grow diamond film with controlled crystalline orientation. The manuscript benefits from a clear motivation and it is an easy and informative read. The manuscript is also excellent in terms of clarity and accuracy of language.

In my view, there are some minor issues that will need to be addressed before becoming suitable for publication:

1: It may be instructive to provide the reader with some comparative studies of the reference experiment where no pre-seeding of the substrate is employed.

2: SEM figure of the deposited nanosheets show not uniform coverage. Can the authors comment how defect density may influence the growth of textured film?

3: caption of Figure 3 is wrong. It is a copy and paste of Figure 2 caption.

Author Response

(The authors gave the same response as above.)

Reviewer 3 Report

It is an interesting paper on the on <110> textured CVD diamond film growth using pre-seeding nanodiamonds. After revision I would recommend it for publication in Materials. However, I have some suggestions/comments/questions:

1. Please add one or two sentence about the diamond nanosheet preparation method and the main idea (ref 19) to the experimental or result section. I think it is an interesting method that worth advertising further. Are you sure that these nanodiamonds are nanosheets? Based on Fig. 1c I would say they have a plate-like morphology. Is Fig. 1c representative? I mean the authors show one AFM image only. I am not sure that a thickness of 10 -60 nm justifies to call a material as “two dimensional”.

2. Please change low and high magnitude to low and high magnification. Fig. 2e is dark and in its current form is not useful. I suggest the authors enhance the visibility of the features shown in Fig. 2e or select a region similar to Fig. 2f and show e.g., diamond spacings.

3. The panel order of Fig. 3 does not match with the fig. caption, also check the text for consitency. Please make it clear in the figure caption that panels a and b correspond to the syringe deposited nanodiamond prepared by the method reported in ref 19 and panels c, d, e and f correspond to the CVD grown nanodiamonds. What is the broad hump at ~ 70-degree 2 theta on Fig. 3e? I do not think it would correspond to the 400 reflection of diamond.

4. I am not sure about the last sentence of the ms. Do the syringe deposited nanodiamonds exhibit a <110> texture? I think a FIB-TEM investigation could clarify the author’s speculation.

Author Response

(The authors gave the same response as above.)

Round 2

Reviewer 1 Report

The paper can now be published